# Geographical Distribution of Three Forest Invasive Beetle Species in Romania

**DOI:** 10.3390/insects13070621

**Published:** 2022-07-12

**Authors:** Nicolai Olenici, Mihai-Leonard Duduman, Ionel Popa, Gabriela Isaia, Marius Paraschiv

**Affiliations:** 1National Institute for Research and Development in Forestry “Marin Drăcea”, Campulung Moldovenesc Station, Calea Bucovinei 73 bis, 725100 Campulung Moldovenesc, Romania; nicolai.olenici@icas.ro (N.O.); popaicas@gmail.com (I.P.); 2Applied Ecology Laboratory, Forestry Faculty, “Ștefan cel Mare” University of Suceava, Universității Street 13, 720229 Suceava, Romania; 3Faculty of Silviculture and Forest Engineering, “Transilvania” University of Brasov, Șirul Beethoven 1, 500123 Brașov, Romania; gabriela.isaia@unitbv.ro; 4National Institute for Research and Development in Forestry “Marin Dracea”, Brasov Station, Closca 13, 500040 Brasov, Romania; marius.paraschiv@icas.ro

**Keywords:** Ips duplicatus, Xylosandrus germanus, *Neoclytus acuminatus*, distribution, Romania

## Abstract

**Simple Summary:**

Biological invasions in the forests of Europe are becoming more frequent, and bark- and wood-boring insects are increasingly important invasive forest pests. Global trade intensification facilitates the introduction and spread of these species across new areas, while climate change and humans contribute to their establishment and spread. If such species enter new territories, the application of appropriate management measures requires—among other things—a better knowledge of their distribution. Using traps with different attractants, we attempted to establish the current distribution in Romania of three of the beetle species that are predicted to be important forest pests: *Ips duplicatus*, *Xylosandrus germanus* and *Neoclytus acuminatus*.

**Abstract:**

*Ips duplicatus* (Sahlberg, 1836), *Xylosandrus germanus* (Blandford, 1894) and *Neoclytus acuminatus* (Fabricius, 1775) are invasive species reported in Romania, but their current distribution is poorly known. The research aim was to provide new information on this issue. A survey was conducted over the period 2015–2017 in 82 locations, using flight-interception traps and bottle traps, baited with different attractants. Data obtained in our other unpublished studies were also taken into account. A total of 35,136 *I. duplicatus* beetles were collected in 30 survey locations. The highest captures were in the log yards of some factories processing logs of Norway spruce (*Picea abies* (L.) H. Karst.). Considering all known records so far, most of these are in the eastern part of Romania, where an outbreak took place during the years 2005–2014, mainly in spruce stands growing outside their natural range. During the survey, 4259 specimens of *X. germanus* were collected in 35 locations, but in our other studies the species was found in 13 additional places. It was collected at altitudes of 18–1200 m, and the largest catches were from beech stands, growing at 450–950 m. *N. acuminatus* was found in only six locations, in the western and southern parts of the country, at low altitudes, in tree stands composed of *Fraxinus excelsior* L., *Quercus* spp. and other broadleaf species, as well as in broadleaf log yards. The results suggest that *I. duplicatus* is established in most parts of the Norway spruce’s range, *X. germanus* is still spreading in the country, with some areas having quite high populations, while *N. acuminatus* is present only in the warmest regions of the country.

## 1. Introduction

Biological invasions are one of the major challenges facing forests today, and their negative effects are becoming more serious, especially in tropical and temperate forests [1,2,3,4,5,6,7]. They are partially facilitated by global climate change but, almost without exception, are mediated by human activity, which helps a huge diversity of organisms to overcome geographical barriers [8,9,10,11].

Various alien organisms have been introduced and are established in forest ecosystems in different parts of the world, with insect species being among the most numerous and having a significant ecological and economic impact [4,5,7,12,13,14].

Although most alien insect species that have established in the forest ecosystems over time are sap feeders and foliage feeders, the rate of introduction of insects that feed on phloem or wood has markedly increased in the last decades [5,15].

In Europe, more than 20 alien species of bark and ambrosia beetles (Coleoptera: Curculionidae: Scolytinae) [15,16,17] and 20 alien species of longhorn beetles (Coleoptera: Cerambycidae) [18,19] are listed. Three of them have also been found in Romania: *Ips duplicatus* (Sahlberg, 1836), *Xylosandrus germanus* (Blandford, 1894) and *Neoclytus acuminatus* (Fabricius, 1775) [20,21,22]. A fourth species, *Trichoferus campestris* (Faldermann, 1835), may also have been established in Romania [23], but we did not find any examples in our study, and consequently it is not discussed in this paper.

*Ips duplicatus*, the northern bark beetle (NBB) or double-spined bark beetle, is a boreal species. Apart from the single specimen used by Eichhoff (1877) to describe the species as *Tomicus infucatus*, which would have been collected in Steiermark during the 19th century, it was found only in the Euro-Siberian taiga [24,25,26]. However, in the first half of the 20th century, it was more frequently found in the territories of the modern-day countries Poland, Czech Republic and Slovakia [27,28,29,30,31,32,33,34,35,36,37,38,39]. During this period, the species was just a faunistic element in Central Europe; it became an economically important pest only in the north-eastern part of Poland, and even there it was less important than *Ips typographus* (Linnaeus, 1758) [40]. This region is regarded as the southern border of the native area of occurrence of *I. duplicatus* [41]. The situation remained apparently unchanged until the early 1990s [42,43], when local outbreaks of this species occurred in the north–eastern territory of the Czech Republic and in the south–western part of Poland [39,44,45].

Since 1989 it has also been repeatedly reported in Austria [46,47,48] and has spread to the whole territory of the Czech Republic and Slovakia [49,50], as well as to other countries. It is currently present in the following European countries: Austria, Belgium, Belarus, Bulgaria, Central and North European Territory of Russia, Croatia, Czech Republic, Estonia, Finland, France, Germany, Hungary, Latvia, Liechtenstein, Lithuania, Norway, Poland, Romania, Serbia, Slovakia, Sweden, Switzerland and Ukraine [51,52,53].

NBB usually inhabits spruces (*Picea* spp.), but when populations reach high densities and primary host trees are missing, pines (*Pinus* spp.) and other conifers may also be affected [27,42,43,54,55]. Consequently, the cultivation of the Norway spruce at low altitudes in the Central European countries supported the expansion of the double-spined bark beetle’s area and the fusion of boreal and alpine ranges [39]. The expansion of NBB was also favoured by the import of un-debarked timber from Russia and Baltic countries, which served as the main entry pathway for this species to Central and Western Europe [39,46,47,48,56,57,58] in the last decades. This introduction pathway was confirmed by genetic analyses [59]. The transportation of infested wood between affected and unaffected zones also favoured the general distribution of the species in some countries [60].

Both in the Czech Republic and Poland, the mass outbreaks of NBB took place in artificial spruce stands located at low altitudes, under 600 m above sea level (a.s.l.), where this bark beetle species has 2–3 generations per year [61,62] and where the spruce trees are severely affected by abiotic factors such as severe drought and heat waves, which are predicted to occur more frequently as a result of climate change [63,64], and biotic factors such as root fungi (*Armillaria* spp.) [45,49,54,65,66]. Although there is a progressive decline in spruce stands, attacks are increasing in the north-eastern part of the Czech Republic and also in other regions [50].

In Romania, *I. duplicatus* was first collected in 1948, in the Eastern Carpathians (Rarău Mountain) close to the border with Ukraine [67]. Later, it was sporadically found in many other locations, mainly in the Eastern Carpathians, at altitudes between 175 m and 1200 m a.s.l. [67,68]. After 2002, when large volumes of infested wood were transported from the mountains to hilly areas, where artificial spruce stands were suffering from drought, an outbreak took place between 2005 and 2014 in the north–eastern part of the country, killing more than 1.3 million trees (0.4 million cubic metres) [21,69].

*Xylosandrus germanus*, the black timber bark beetle or black stem borer (BSB), originated in South-East Asia [15]. It was first reported in Europe by Groschke [70], who found this species in 1952 near Darmstadt, Germany. However, Wichmann [71] suggested that it could have been introduced from Japan, with oak wood, during the years 1907–1914 and 1919–1929. The introduction from Japan, most likely from Honshu, was demonstrated by Dzurenko et al. [72], who suggested that it happened before the Second World War.

From Germany, BSB spread rapidly across Europe, especially after 2000, and is now reported in Austria, Belgium, Croatia, the Czech Republic, Denmark, France, Germany, Georgia, Hungary, Italy, Netherlands, Romania, Slovenia, Spain, the South European Territory of Russia, Sweden, Switzerland, Turkey, Ukraine and the UK [51,73].

This species is an ambrosia beetle, feeding on the fungus *Ambrosiella grosmanniae* C. Mayers, McNew & T.C. Harr., which is cultivated by females inside the galleries [74]. Consequently, it does not depend on the tree species itself but on the temperature and the humidity, which ensure fungus growth. It is a very polyphagous species, colonizing mainly broadleaves but also conifers [75]. Usually, it attacks dying or recently dead trees, stumps and logs [42], but can also attack apparently healthy trees, which release ethanol after exposure to stress factors (drought, flood, freezing, etc.) [76,77,78,79].

Although considered a secondary pest in Europe [42], in recent decades, with the expansion of its geographical range, probably exacerbated by climate change, it has caused damage both in the forest industry, by colonizing logs of beech, oak, Norway spruce and silver fir [80,81], and in young plantations of fruit trees [82,83] and grape yards [84,85].

In Romania, BSB was first collected in 2009, in an uneven-aged mixed forest of European beech (*Fagus sylvatica* L.) and sessile oak (*Quercus petraea* Liebl.) located in the western part of the country, and then in 2011–2014 in old-growth European beech and mixed (beech and conifers) forests in the northern and central parts of the country [22,86].

*Neoclytus acuminatus*, the red–headed ash borer (RHAB), is a North American species [87] that was first collected in Europe in the mid-19th century in Rijeka (then Fiume), Croatia [88]. From that place, it spread west and east to other countries [89], but there were also other repeated introductions from North America into England, Germany and Italy with ash (*Fraxinus* spp.) wood [90]. It has been reported in Austria, Bosnia and Herzegovina, Croatia, the Czech Republic, France, Germany, Hungary, Northern Ireland, Portugal, Serbia and Montenegro, Slovakia, Slovenia, Switzerland and the United Kingdom [18,89,91,92], with established populations in Croatia, Italy, Hungary and Montenegro [89]. RHAB is a highly polyphagous species, found mainly on broadleaves, with ash trees (*Fraxinus* spp.), oak trees (*Quercus* spp.) and hackberry (*Celtis occidentalis* L.) being the preferred hosts in its native range [87,93].

The insects attack weakened, dying or recently dead trees, as well as recently cut logs with bark. The larvae develop in the inner bark and sapwood of the trunk and branches. The attacked young trees become prone to wind breakage, because the larvae gnaw tunnels vertically along the trunk, as well as around the circumference. When infestation is high, the wood quality of the logs is severely reduced [87].

RHAB is one of the most common wood borers throughout the eastern United States and in south-eastern Canada, and it heavily attacks fresh logs of ash, oak and hickory [87,94]. In Europe, there has been no significant damage reported so far, but it is considered to be a potentially important pest of *Celtis australis* L. in Slovenia, Croatia and Hungary [95].

In Romania, prior to our research, only one RHAB individual had been collected, in 2002 [20], and therefore it was not known whether the species was already established or whether this was a simple interception of a specimen arriving at Timisoara from south-eastern Hungary, where it was already relatively common in the late 1990s [96].

As they are secondary pests, which mainly attack weakened, dying or recently fallen trees, it is expected that NBB, BSB and RHAB, like other pest species, will be favoured by current climate change, which will amplify the frequency and magnitude of natural disturbances and will weaken the defences of trees [97,98,99,100,101,102,103,104,105].

In addition, these species could be very easily transported over long distances through trading infested wood at the regional, national or international scale [106,107,108], because they live for quite a long time under the bark or in the wood.

In this context, it is very important to take into account the presence of new pests both in the management of forests and in the harvesting, transportation, processing and trading of wood products such as timber, fuel wood, dunnage, wood for packaging, etc. [49,109,110]. For this purpose, knowledge of the distribution of these species in the country is needed. Consequently, the purpose of our study was to obtain information on the distribution of these species in Romania and to highlight the contribution of human activities to their introduction, establishment and spread at a national level.

## 2. Materials and Methods

### 2.1. Study Areas

Knowing that the presence of the three species (NBB, BSB and RHAB) has already been reported in Romania, extensive field research was carried out in 82 locations between 2015 and 2017 to determine their distribution throughout the country (Figure 1). The research was also aimed at detecting any other alien scolytid and cerambycid species whose presence had not been reported previously. The study locations were chosen randomly, taking into account mainly the ecological requirements of the targeted insect species, in order to increase the chances of detection (capture). At the same time, we also attempted to highlight the possible influences of human activities (mainly the expansion of softwoods, especially spruce, outside their natural ranges and the transportation of infested wood from affected stands in other parts of the country) on the distribution of these species in the country.

In 2015, the study focused on BSB. The presence of this species was verified in 13 locations, situated in mountain or hilly areas, at altitudes between 200 m and 1340 m a.s.l., in mixed or broadleaf forests, most of them aged 70–110 years, composed of native tree species. *Fagus sylvatica* L. was present in all locations, in pure stands or mixed with coniferous trees (*Picea abies* (L.) H. Karst., *Abies alba* Mill., *Larix decidua* Mill.) or other deciduous trees (*Quercus petraea* (Matt.) Liebl., *Carpinus betulus* L., *Tilia cordata* Mill., etc.).

In 2016 and 2017, the study was extended to the detection of invasive beetles belonging to the Scolytinae subfamily and Cerambycidae family. Thus, field observations were conducted in 69 new locations (53 in 2016 and 16 in 2017), covering almost the entire country’s territory, between sea level and 950 m a.s.l., predominantly in the hilly and plain areas. Most locations (43) were situated in tree stands (coniferous, deciduous or mixed forests) that were 25–135 years old. The tree stands were mainly composed of native tree species (*P. abies*, *A. alba*, *L. decidua*, *Pinus nigra* Arn., *Acer campestre* L., *A. platanoides* L., *A. pseudoplatanus* L., *Alnus glutinosa* L., *Betula pendula* Roth, *C. betulus*, *Fraxinus excelsior* L., *F. ornus* L., *F. sylvatica*, *Prunus avium* L., *Quercus cerris* L, *Q. fraineto* Ten., *Q. petraea*, *Q. pubescens* Willd., *Q. robur*, *Populus tremula* L., *T. cordata*), with fewer exotic species (*Acer negundo* L., *Pseudotsuga menziesii* var. *glauca* (Schwer.), *Robinia pseudoacacia* L.). Another 26 locations were chosen outside the forests, in places where alien beetle species could be detected (near wood processing factories, ornamental plant stores, wood yards, the vicinity of international airports, a seaport and customs areas). Descriptive information for each location is provided in Appendix A.

### 2.2. Insect Sampling

The studied insect species (NBB, BSB and RHAB) were caught using flight-interception traps similar to Intercept™ Panel Traps (PT) and bottle traps (BT). PTs were chosen because they are effective in capturing bark beetles and longhorn beetles [111,112,113], while BTs are especially effective in capturing *X. germanus* beetles [114,115]. The bottle traps used in the study were constructed of two clear plastic 1.5 L bottles with the mouths connected by a plastic tube. The upper bottles had three vertical slits of 11 × 3 cm to allow the entrance of insects. The lure was hung within the upper bottle, at the level of the openings. The lower bottle served as a collection jar. All the traps, except those used in 2015, were wet traps, with collection cups containing water with salt (sodium chloride) and traces of detergent.

The traps were baited with synthetic pheromones or kairomones (Table 1 and Appendix A). AtraDUP^®^ pheromone lures, containing the main pheromone components of *I. duplicatus*, Ipsdienol and E-myrcenol [116,117,118], were used for the collection of NBBs. Ethanol was chosen as an attractant for *X. germanus*, because the females of this species respond quite well to alcohol [114,115,119]. A combination of ethanol and (-)-alpha-pinene, released from two separate dispensers, was used to attract the adults of *N. acuminatus*. This combination had previously allowed the detection of this species in Slovenia [120,121]. AtraTYP^®^ and AtraLINEA^®^ pheromone lures, normally used to attract *I. typographus* and *Trypodendron lineatum* (Olivier, 1795), respectively, have been deployed to attract other alien scolytid beetles that we expected to find during the survey. All lure types and panel traps were provided by the Raluca Rîpan Institute for Research in Chemistry, Cluj-Napoca.

In the field, the traps were placed in groups, with three to five traps in each location (Table 2), at a distance of 50–100 m from each other. In the coniferous and mixed forests, the traps baited with pheromone lures for bark beetles were set up in the open at a distance of 10–15 m from the tree stand’s edge, while the other traps were placed at the stand’s edge or 5–10 m within the tree stand, but at a distance of at least 2 m from the nearest tree. The PTs used to collect bark beetles and longhorn beetles were installed with the mouth of the collecting funnel at a height of 1 m above the ground, as in other studies concerning these species [121,122,123,124], while those used for BSB were placed 0.5 m closer to the ground, and the BTs were placed with the openings 0.5 m from the ground, because these beetles fly near the ground [114,125].

In 2015, the field work was conducted over 3–6 weeks, from mid-June to mid-August, while in 2016 the traps were installed in the field between 5 May and 10 June, and insect collection lasted 12–20 weeks. The field work started much earlier in 2017, between 28 March and 27 April in most places, and the survey was generally stopped after 16–21 weeks, except in one place where it lasted only 8 weeks. We collected insects from traps every 2–3 weeks, and replaced lures when necessary, depending on field life. After collection, the biological material was conserved in 70% ethanol and kept cold (at −20 °C) until it was analysed.

### 2.3. Additional Data concerning NBB, BSB and RHAB Presence

Some additional data on the presence of NBB and BSB were collected within several field experiments we conducted during the years 2009–2018 in different locations (Appendix A). Other additional data concerning the presence of NBB are taken from our unpublished studies on the NBB outbreak in the north-eastern region of Romania (Appendix A). Historical data concerning the presence of NBB in the country were assembled by searching the information available in the literature [21,126].

In order to obtain a full updated picture of RHAB’s current distribution in Romania, we also took recently published data into account [127,128].

### 2.4. Species Identification

The taxonomical identification of the collected specimens at species level was performed by the first author using the latest identification keys, including those of Pfeffer (1995) [43] for the Scolytinae subfamily and Bense (1995) [129] for the Cerambycidae family.

### 2.5. Climatic Data

To interpret the spread and current distribution of the three species in Romania, we used a series of climatic indicators for the period 1950–2017 for every study location: mean annual temperature (MAT), mean annual precipitation (MAP), de Martonne aridity index (IdM), minimum temperature in January (_Tmin-Jan_) and the minimum, mean and maximum temperatures of each season (winter (DJF), spring (MAM), summer (JJA) (T_min-S_, T_mean-S_, T_max-S_) and autumn (SON)). Climatic data were derived from the daily gridded dataset E-OBS, with a resolution of 0.25° × 0.25° [130].

### 2.6. Wood Import Data

The distribution of BSB and its very rapid spread in the country suggests that the introduction took place mainly via the importation of infested wood. Since there is no interception report published on this issue by the authorities, we attempted to verify this hypothesis by analysing data concerning Romania’s round-wood imports since 1997, from all regions where BSB was present (South-East Asia, North America and Europe). According to the FAO (2018), Romania imported a volume of 8,819,977 m^3^ between 1997 and 2016, with 99.9% coming from Europe. From all European partners, we selected only those countries from which more than 50,000 m^3^ were imported in these 20 years.

### 2.7. Data Analyses

Since the main objective of the research was to determine the current distribution of species in the territory, all catches of the three species were taken into account, regardless of the attractants with which the traps were baited (Appendix A). However, for analysing the relationship between NBB and BSB catches and altitude, only insects from traps baited with AtraDUP^®^ and ethanol, respectively, were taken into account. If no individuals of NBB and BSB were captured in the specifically baited traps, those locations were removed from the analysis, even if there were infrequent catches in traps baited with other attractants. For this analysis, data from other studies conducted during a whole season (Appendix A), e.g., 2016–2017, were also taken into account. The total number of BSB individuals collected in 2015 was divided by three, because in that year three ethanol-baited traps were placed in each location.

The presence of increasing or decreasing trends in seasonal climatic data was tested using the non-parametric Mann–Kendall trend test Z statistic [131]. Positive values of Z indicate increasing trends, while negative Z values show decreasing trends. If Z is lower than the theoretical value at the significance level (*p* = 0.05), then no trend is observed. Statistical analysis was carried out using the trend package in the R environment (version 4.1.0) [132].

### 2.8. Maps

All distribution maps were generated in the ArcMap 10.2.2 software (Esri, Redlands, CA, USA), using the STEREO 70 projection system based on the “Dealu Piscului” Datum (EPSG: 31700). The layer of forest areas was generated using Corine Land Cover (CLC) 2012, Version 18.5.1 (European Environment Agency, 2016), while the layer of climatic influences was extracted from the map “Climate regions and topoclimates” [133]. The map of average summer temperatures in Europe was obtained from the Atlas of the Biosphere [134].

## 3. Results

### 3.1. Ips duplicatus

During 2016–2017, a total of 35,136 NBB individuals were captured (Appendix A), most of them (77.3%) with pheromone traps baited with AtraDUP^®^ and 22.3% with pheromone traps baited with AtraTYP^®^. The species was collected in only 30 of 52 locations, mainly in conifer and mixed species stands or within wood yards, but also in broadleaf stands and in places without forest vegetation (Table 3, Figure 2). More than half of the captures (53.6%) were from the six log yards for conifer timber (Dumbrava Roșie, Brașov_2, Sebeș, Vermești, Reci, Reghin) and 41.9% from the five most infested conifer and mixed stands (Piatra Șoimului, Hemeiuș, Borsec, Lunca Bradului, Praid), located in the eastern and central parts of the country, while in the remaining places the captures totalled less than 5% (Figure 2 and Figure 3).

Most of the 30 locations were in or close to the Carpathian Mountains, where previous studies had documented the presence of this species, but some of them were far from the mountains in the hilly or plain areas of the country, and one of them was even on the shore of the Black Sea (Figure 4). In such places, located at low altitudes (<300 m), the numbers of captures were very low (≤10 beetles/location/season), except in the cases of log yards.

Taking into account all existing investigations on this species, most records are from the north-eastern part of the country, where the climate is influenced by Baltic and eastern continental air masses. In this region, NBB developed active foci between 2005 and 2014 (Appendix A). This explains the higher values of the catches presented as “other own data” in Figure 5.

The sizes of captures in the forests, during the years 2016–2017 and also in other years (Appendix A), varied with tree stand composition (Figure 3) but also with altitude (Figure 5), and the data suggest a higher level of populations at 350–600 m a.s.l., where Norway spruce was planted 50–65 years ago, outside its natural range.

### 3.2. Xylosandrus germanus

During the study conducted between 2015 and 2017, a total of 4259 BSB adults were collected, with 835 collected in 2015 using dry PTs baited with ethanol and 3424 in 2016–2017, when wet PTs and BTs baited with different attractant types were used (Appendix A). In 2016–2017, 80% of BSB captures were from traps baited with ethanol, and 9.6% and 9.7% from traps primed with ethanol plus alpha-pinene and AtraLINEA^®^, respectively, while the remaining proportion was shared equally between traps baited with AtraDUP^®^ and AtraTYP^®^.

The species was collected in 35 of the 79 monitored sites, especially in broadleaf and mixed tree stands, but also in conifer stands, log yards and other places (Table 4). However, the captures were very low (≤10 beetles/trap/season) in most locations (Figure 6), and substantial captures (>100 beetles/trap/season) were found in only seven places, especially within stands with more than 50% beech in their composition (Figure 7). In contrast to the findings in the case of the previous species, very few specimens of BSB were captured in wood yards. Beetles captured in wood yards represented only a very small part (0.8%) of the total captures, and most of them were collected in the wood yard at Brașov_2.

Like NBB, BSB was mainly found in and close to the Carpathian Mountains, in forested areas where the average temperatures in the summer months fall within the range of 16–21 °C. BSB was found in the lowland area and in the floodplain of the Danube in only a few cases and was not collected at all in the most arid regions of the country (Figure 8). Consequently, although BSB catches were recorded from 18 m to 1185 m a.s.l., most of them, both in this study and in our other studies (Appendix A), were found at altitudes of 450–920 m (Figure 9). In the case of this species, the higher catches from other studies conducted by us were due to the intentional choice of places with a higher level of populations, to achieve the specific objectives of those studies.

### 3.3. Neoclytus acuminatus

Only 15 RHAB specimens were collected during the years 2016–2017, from only six places. Ten specimens were caught with pheromone traps primed with AtraLINEA^®^, three with ethanol and only two with ethanol plus alpha-pinene. All captures were from places located at very low altitudes (15–220 m); two in the western part of the country (with oceanic climatic influences), three in the south-western part of Romania (with sub-Mediterranean influences) and one in the southern part (with transition climatic influences between sub-Mediterranean and continental) (Figure 10). In those locations, the average annual temperature is 9.5–12 °C, while the average summer temperature is 19.0–22.8 °C (Appendix A).

The species was collected within three tree stands which had different oak species (*Q. cerris* L., *Q. frainetto* Ten. and *Q. petraea* (Matt.) Liebl.) in their composition, within two wood yards where there were logs of different broadleaf species, including *Fraxinus* sp. and *Quercus* sp. and within one locality (Dorobanți, near Curtici) with scattered trees. Most of the captures (10) were recorded in the wood yard at Oltenița, near the Danube River.

## 4. Discussion

### 4.1. Ips duplicatus

As expected, NBB beetles were captured not only in the Eastern and Southern Carpathians and in the neighbouring areas, where the species had been repeatedly collected before, but also on the Transylvanian Plateau and in the Western Romanian Carpathians, the Western Hills and Getic Subcarpathians. However, the highest numbers of captures were from the Eastern Carpathians and Moldavian Subcarpathians, suggesting that the level of the NBB population in this zone is higher than in the rest of the country.

The higher NBB populations in the sub-mountainous areas of the Eastern Carpathians, Moldavian Subcarpathians and Moldavian Plateau is the result of a complex of factors that acted synergistically to support beetles’ multiplication. First of all, in the northern part of Romanian Eastern Carpathians (REC), the species has been present for the longest period, being first collected here in 1948, in the natural spruce area [67]. It probably arrived through the gradual dispersal of insects from Central Europe (Czech Republic, Slovakia) along the Carpathians [126] and remained as just a faunistic element until 2000 [21].

Secondly, in the northern part of the REC, the spruce (the main NBB host) is the dominant species in the forests, even in the belt of mixed forests (spruce, fir and beech), because its proportion was artificially increased to the detriment of the other species in the second part of the 19th century and during the early part of the 20th century [135]. In the second half of the 20th century, especially between 1960 and 1985, spruce was planted in the hilly areas, at altitudes of about 300 m in the north of the country and 500 m in the south and west [136,137]. In this way, spruce, a species that optimally grows at annual average temperatures of 4–7 °C and annual precipitation of 800–1200 mm [138], has reached places located far beyond the eastern boundary of its area at Romania’s latitude [139], where average annual temperatures are 8–9 °C and average annual rainfall is under 600 m [136] (Appendix A gives values computed for the years 1950–1983).

The spread of NBB across the country, and especially in the eastern part, occurred in conditions where the average annual and seasonal temperatures were generally experiencing an upward trend after 1984, while the rainfall remained relatively constant (Appendix A).

Due to recent climate change, which has been felt more strongly in north-eastern Romania than in other areas where spruce has been planted outside of its natural area [140,141,142], many of these forest stands are now in areas with annual temperatures above 9 °C, annual rainfall below 550 mm and an annual de Martonne aridity index (IdM) lower than 30 [143] (Appendix A gives data for the years 1984–2017). According to Botzan et al. [144], such IdM values are characteristic of steppe rather than forest zones.

Since spruce has low drought tolerance and requires an adequate supply of soil moisture [145,146], the trees have been affected by water scarcity under the new circumstances. The situation has become more and more serious with the aging of the trees and the increase in water demand, as many of these spruce stands have not been properly thinned and have more than 2000 trees/hectare after 40–50 years [69,137]. Such dense stands have higher transpiration rates and higher water losses due to rain interception in the canopy than thinned tree stands [147].

Site conditions like those in places at low altitudes where spruce has been artificially introduced, adversely affect the physiological functioning of spruce trees and diminish their ability to defend against bark beetles and other harmful organisms, especially in drought years [148,149,150,151,152,153], which seem to have become increasingly common in this area in recent decades [154]. Consequently, during the years 2011–2012, when the Standardized Precipitation Index (SPI–12) averaged at country level was between −1.5 and −1.0 [155], we noted that spruce trees at low altitudes were so dehydrated that they did not release any resin when the beetles penetrated the bark. This could explain why even the large trees (with DBH over 30–40 cm) in sub-mountain areas were almost exclusively colonized by *I. duplicatus* and not by *I. typographus*, as such severely stressed trees are less attractive to the eight-toothed bark beetle [152].

In this context, on 6–7 March 2002, a strong storm hit the REC forests, affecting trees amounting to 7.6 million cubic metres in an area of 539 thousand hectares. The most severely affected were the forests in the county of Suceava: 6 million cubic metres on 256 thousand hectares. Since 2003, in the windthrow-affected area, a large-scale outbreak of bark beetles took place, and more than 0.5 million standing trees were attacked [156]. The trees killed by wind and bark beetles trees were logged between 2002 and 2005, and considerable quantities of bark-beetle-infested timber reached the hill area of the counties in the eastern part of the country, thus increasing the populations of bark beetles considerably, including NBB, in the zones with spruce stands outside the natural range.

The first foci of bark beetles in this area occurred in old spruce forests (80–100 years), where *I. typographus* was the dominant species, such as in the mountain area. Furthermore, attacks began in the young spruce stands (30–50 years), where the dominant species was *I. duplicatus*. One possible explanation for the NBB becoming a dominant species could be that it escaped *I. typographus*’s competition, because the later species cannot colonize trees with bark thinner than 2.5 mm [157], and the young trees, grown in dense stands, had a relatively thin bark.

At low altitudes, the climate not only weakened the vigour of spruce trees but also favoured the faster development of beetles. In such areas, NBB could produce 2–3 generations per year [61,158]. Consequently, the spruce stands at low altitudes were partially decimated by bark beetles [69], and in recent years the tendency of attacks to advance to higher altitudes has become increasingly apparent. This may well be due to the global warming of the last decades, which is felt more strongly at low and middle elevations of the mountain area, especially in spring and summer, and is associated locally with a reduction in precipitation in the winter and spring and increased climatic water deficit on the eastern side of the Eastern Carpathians and on the Moldavian Plateau [142,159].

In 2018, two NBB foci were found in the inner part of the REC, one at Crucea (45.352400° N, 25.615726° E, 738 m a.s.l.) and one at Valea Putnei (47.479208° N, 25.380858° E, 1000 m a.s.l.). However, each focus was not far (less than 500 m) from a wood yard, and the concentration of infested logs in these yards could have facilitated the local increase in the NBB population. High levels of NBB populations at lower altitudes (up to 600–800 m a.s.l.), were mentioned also in the Czech Republic, Poland and Slovakia [50,160,161].

In the other regions of the country, especially in the west and south, the capture numbers suggest a much lower level of NBB populations; in some locations, they could be only the result of the interception of passing insects, without established populations in those areas. However, the situation may change as a result of infested wood transportation from tree stands to other areas, as revealed by the very large catches in wood yards, including that located in Sebeș. The same conclusion is suggested by catches from Constanța seaport, as well as from the areas around Timișoara and Otopeni airports, which are places located far away from any spruce tree stands.

Furthermore, more intensive research should clarify whether the captures from western and southern parts of the country correspond to established populations or not, because the climatic characteristics of those zones are not an impediment to the survival and development of this species. At any rate, given increasing global warming and considering the magnitude of the attacks during the 2005–2014 outbreak, NBB should be considered an increasingly important pest of Norway spruce in Romania, especially in the hilly and sub–mountainous areas. Consequently, to avoid damage caused by NBB, forest managers should take into account both climate change [162,163] and the presence of this species.

### 4.2. Xylosandrus germanus

Although it was collected for the first time in Romania only a decade ago (in 2009), this species was found in 2015–2017 in many places, spread throughout the country. Of these places, two locations (Groșii Noi and Voievodeasa), situated more than 320 km apart, were identified as early as 2009 and 2011. As a result, although the species spreads at a rate of tens of kilometres per year [164], it is supposed that its current distribution in Romania is the result of several introductions in different places in the country. In addition, it is very likely that the species arrived in the country some years before it was detected, as it has been reported in Hungary since 2005 [165]. A similar situation was observed in Slovakia, where the species was first reported in 2010 [73]. It is very likely that the species arrived for the first time in Romania during the period of rapid increase in wood imports after the year 2004, when tens of thousands of cubic metres of round wood were imported from countries where the pest was already present (Appendix A).

The very large variability in catches suggests the existence of populations with varied densities throughout the country, with the largest catches at altitudes ranging from 450 m to 950 m, in stands with at least 50% beech in their composition. These data confirm that the species finds optimal development conditions in beech forests [73]. In addition, in Belgium, the species is widespread and has reached high population levels, especially in mixed hardwood forests dominated by beech [164], but at much lower altitudes than in Slovakia and Romania. Other relatively large populations, which can be considered as established, have been identified at altitudes between 250 m and 450 m, both in the eastern part of the country (Roznov, Valea Budului, Sărata) and in the south (Tismana) in the Subcarpathian area. However, in most study sites located at an altitude of under 450 m, no or very few insects were captured.

In the survey conducted during 2015–2017, the largest catch (1058 beetles/trap/season) was recorded in Braşov, but similar and even larger catches (up to 7426 beetles/trap/season) were obtained in our other studies conducted in the northern part of the Romanian Eastern Carpathians, at Cacica, Voievodeasa and Palma (Appendix A). In each of these two regions, there is a large industrial woodworking unit, which is supplied with wood not only from the country but also from imports, and several BSB specimens were captured in the log yard of the Braşov wood processing factory (Braşov_2 in Appendix A). This could mean that log yards play the role of hubs in the spread of BSB in the country.

On the other hand, BSB was captured in only 5 of the 12 wood yards in which ethanol-baited traps were installed (Table 4 and Appendix A), and the number of catches was very low at each site (1–29 specimens) (Appendix A), a fact which may suggest that the spread of this species in the country is facilitated to a lesser extent by the transport of infested wood, compared to NBB, since no log or firewood infestations with BSB have been reported so far in Romania. However, it must be kept in mind that the results we present are also determined by the biology of this species. BSB females remain in the colonized wood throughout the development of the new generation and, as a result, can no longer be intercepted in flight. In addition, in areas where the species has only one generation per year, the young beetles emerge from the substrate in which they developed only in the following spring [75] and can be captured during the flight period only if wood is not processed in the meantime or the insects are not destroyed during processing.

In the case of this species, a single viable female is sufficient to lay the foundations of a new population, because mating takes place almost exclusively before leaving the galleries in which the beetles developed [166,167], they can develop in a large number of wood species [125], and colonization of the substrate does not require the concentration of a large number of individuals, as in the case of bark beetles attacking seemingly healthy trees.

Very small catches recorded in some low-altitude localities could mean a much lower population level due to a too-dry climate, as observed in Slovakia [73], but at the same time, it could be the result of the very low sampling effort (only one ethanol-baited trap per location) or of installing the traps too late (in 2015). However, the lack of catches in many places in the western part of the country (where BSB was first collected in Romania) is problematic and requires more careful sampling, as it is known that the number of catches depends greatly on the habitat where the traps are located and their distance from the ground [78,114,168]. The species prefers shady sites within the forest [169,170], and the beetles fly near the ground [171,172].

In the western and central parts of Europe, permanent populations of this species have been recorded only at altitudes below 600 m [164,173,174,175,176], with the sole exception of Slovakia, where spruce logs were attacked by *X. germanus* at an altitude of 1020 m [73]. In Romania, the species was collected in four different places at altitudes above 1100 m (in Bădeanca at1185 m, in Bobeica at 1195 m, in Cârlibaba at 1270 m and in Căpăţâneni at 1520 m). In all four cases, only one specimen was captured, and it is thought that these could be isolated specimens entrained by upward air currents from lower altitudes and from relatively large distances. Additional investigations are needed to determine the extent to which the species is present in the inner part of the mountains, where spruce predominates, and up to what altitude this species can survive in the mountain range. However, it is certain that at the periphery of the Carpathian chain there are large populations at altitudes of 850–900 m in the northern part of the Romanian Eastern Carpathians and at altitudes of more than 900 m in the Southern Carpathians.

It has been suggested that species spreading at high altitudes is limited by minimum winter temperatures [164,175]. However, in the Voievodeasa forest reserve, where the presence of the species was documented in 2011, no reduction in 2012 catches was noted [22] after the frosty weather that affected all of Eastern Europe in January and February 2012. At that place, the average daily minimum temperature in the period 25 January to 16 February 2012 was −17.4 °C, and on February 2nd and 3rd, the daily minimum temperatures dropped to −25.8 °C and −25.0 °C, respectively [130,177].

It is likely that the apparently negligible effect of the frost on the insect population was due to the protection afforded by the snow on the ground, since this species usually colonizes appropriate breeding material lying on or in the soil, and the standing trees are mostly colonized at the base of the stem [75,178,179]. According to the WMO [177], in the Voievodeasa area, the maximum snow depth in the period 1 December 2011–29 February 2012 was between 21 cm and 50 cm, and the number of days with snow on the ground in the same period was between 61 and 80. According to these observations, negative temperatures of up to −25 °C do not necessarily seem to affect the survival of beetles of this species if there is a thick layer of snow in the area during frost. Similar results have been reported in Slovakia by Dzurenko et al. [180].

On the other hand, the data suggest no positive trend of minimum temperatures during the winter in the last 34 years (Appendix A). Consequently, the extension of the BSB-infested area to higher altitudes is not the result of rising winter temperatures but of other factors.

An alternative explanation for the limited spreading of species at high altitudes might be insufficient warmth in summer. It is known that the optimal temperature for the development of *X. germanus* insects is 21–23 °C [181], and this overlaps with the optimal temperature range for the BSB’s symbiotic fungus, which is 20–25 °C [182]. In addition, if the maximum daily temperature does not exceed 20–21 °C for at least two consecutive days, the beetles do not fly and do not colonize the available substrate [76]. As a result, areas where there are fewer days per year with maximum temperatures above 20 °C may be considered less favourable for colonization by *X. germanus*, regardless of altitude, while in areas with daily maximum temperatures that are optimal for the development of insects throughout the summer, large population growth is expected. Thus, in the places where we found the largest BSB populations, in Brașov and Cacica, and in Palma and Voievodeasa, the average daily maximum temperatures in the summer (T_max-S_) were 23.1 °C and 22.1 °C, respectively (Appendix A).

Consequently, it is expected that the spread of BSB will extend to higher altitudes due to global warming that is felt mainly during the spring and summer months (Appendix A), especially in the Eastern Carpathians and the Apuseni Mountains [159].

In areas with optimal conditions for population growth, it is expected to become an important pest, both in forests and in orchards, as mentioned in the Introduction, especially where the orchards are located near forests. The attack on fruit trees in an orchard in Vileacu de Beiuş in the vegetation season of 2018 (Ecaterina Fodor, Oradea University, personal communication) is the first reported case in Romania.

### 4.3. Neoclytus acuminatus

During our study, this species was collected in only six locations and—apart from one location—only one specimen was captured in each place, suggesting that the species is relatively scarce in Romania and that populations are very low or there is still no established population within the country. However, the actual distribution of RHAB in Romania may be much greater than suggested by our trapping survey data, because the lures used (ethanol, alpha-pinene) are much less effective than the species aggregation pheromone syn-2,3-hexanediols [183,184]. For example, in Georgia, USA, funnel traps baited with a combination of syn-2,3-hexanediols plus ethanol had mean captures of about 100 RHAB individuals, compared to zero or nearly zero catches in traps baited with ethanol alone [185]. In addition, in nine of the western and southern locations, the traps were baited by mistake with alpha-pinene only. Moreover, trap checking and replacement of alpha-pinene dispensers was conducted only at intervals of 2–3 weeks, despite the fact that operators who worked in the field were warned that alpha-pinene dispensers do not function for longer than a week. Practically, those traps were inoperable for at least 50% of the time. This explains why only two of the captured specimens were found in traps baited with ethanol and alpha-pinene.

The type of trap and the locations of traps in the field, both in relation to the edge of the forest and the height from the ground, could also have had a negative impact on the size of the catches. Experiments performed in other countries [186] have shown that multiple funnel traps are much more effective than panel traps in capturing *N. acuminatus* specimens, and Fluon–treated panel traps are 10 times more effective than untreated panel traps such as those used in this study. In addition, when traps were placed with the mouth of the collecting funnel at about 1 m from the ground, those located outside the forest or at the forest’s edge captured considerably fewer specimens than those inside the forest [187].

Recently, Rassati et al. [188] showed that *N. acuminatus* adults are much more frequently captured in traps placed in the tree crowns than in those under the canopy close to the ground, probably because the adults of this species prefer to lay their eggs in the cracks in the bark of the branches from the crown (more than 10 m from the ground) and not on those on the soil [189]. However, Miller et al. [184] captured more *N. acuminatus* in traps placed at a height of 1.5 m than in traps placed in the canopy.

Taking into account all the above-mentioned aspects, the small numbers of detections and catches for this species in our survey is understandable. The fact that the species was found in the western and south-western parts of the country suggests that it probably came to Romania from Hungary, where it has been relatively common since the 1990s [96]. However, it is now also present in the eastern part of the country, with apparently established populations [128], and this fact could be due to other introductions. For now, its distribution seems to be linked to forests where the preferred host species (mainly *Fraxinus* and *Quercus*) are present. These sites are also characterized by a warmer climate. Due to the warm climate in the infested area, insects are likely to end their development in just one year, as in southern Europe [95]. The presence of the species in wood yards suggests that the transport of infested wood contributes to the spread of the species in the territory. Since this species attacks weakened or dying ash trees, it is supposed that it will benefit from ash dieback caused by *Hymenoscyphus fraxineus* (T. Kowalski) Baral, Queloz & Hosoya, which is progressing from the east of the country to the west [190]. It could also become a major pest of the black locust, as in North America it has been reported as attacking seemingly healthy trees of this species [87].

## 5. Conclusions

The data we have considered so far suggest that NBB is present in most of the spruce-growing areas in Romania and that it is expected to become more damaging as the global climate changes, impacting both host trees and local entomofauna.

BSB is already largely present in the country and continues to spread along with population growth in areas with permanent populations. It has already begun to appear as a pest in orchards, and this is expected to become more frequent.

RHAB is widespread in the western, south-western, southern and eastern parts of the country at low altitudes, but it is expected that its distribution area will continue to expand, because forests with oak and ash species in their composition are becoming more widespread.

Although our evidence is indirect, we consider that human activities (the expansion of spruce outside its natural range, wood imports, domestic trade in infested wood) and climate change in recent decades have contributed—to a greater or lesser extent, depending on the species considered—to the introduction, establishment and spread of these three species in our country.

In order to avoid the damage caused by these pests, those who manage the forests and orchards of the country should take into account the presence of these new species. The present study provides information on their distribution in Romania and the context that led to the current situation. Past events should be lessons for better management in the future.

## Figures and Tables

**Figure 1 insects-13-00621-f001:**
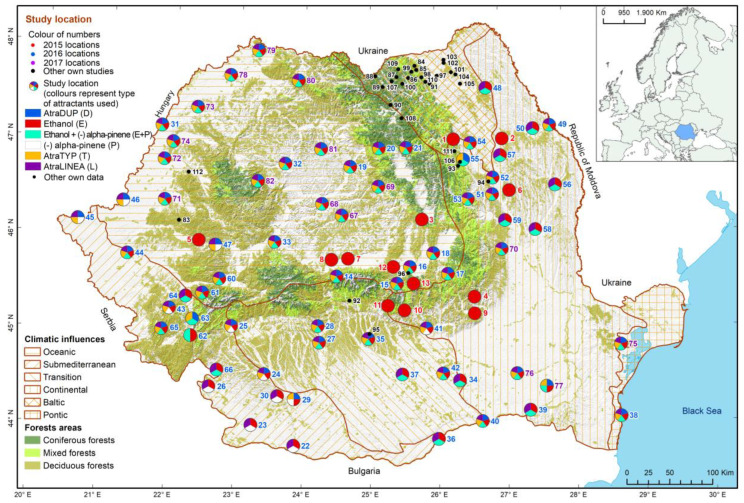
Location of study sites in the survey conducted during 2015–2017, as well as those of our other studies that allowed the collection of NBB and BSB specimens. (The correspondences between point numbers and names of locations are given in Appendix A).

**Figure 2 insects-13-00621-f002:**
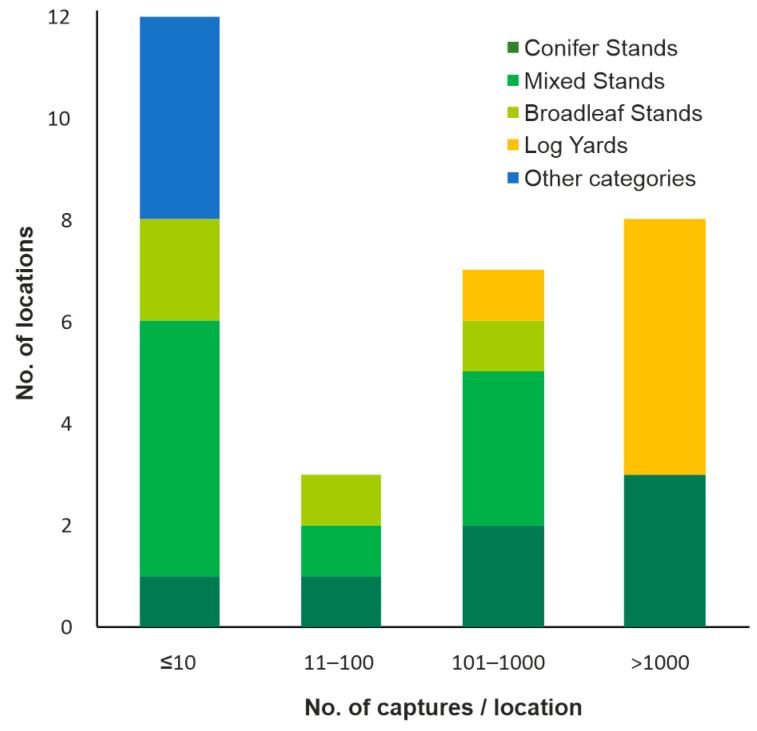
Number of locations and habitats with different NBB capture sizes.

**Figure 3 insects-13-00621-f003:**
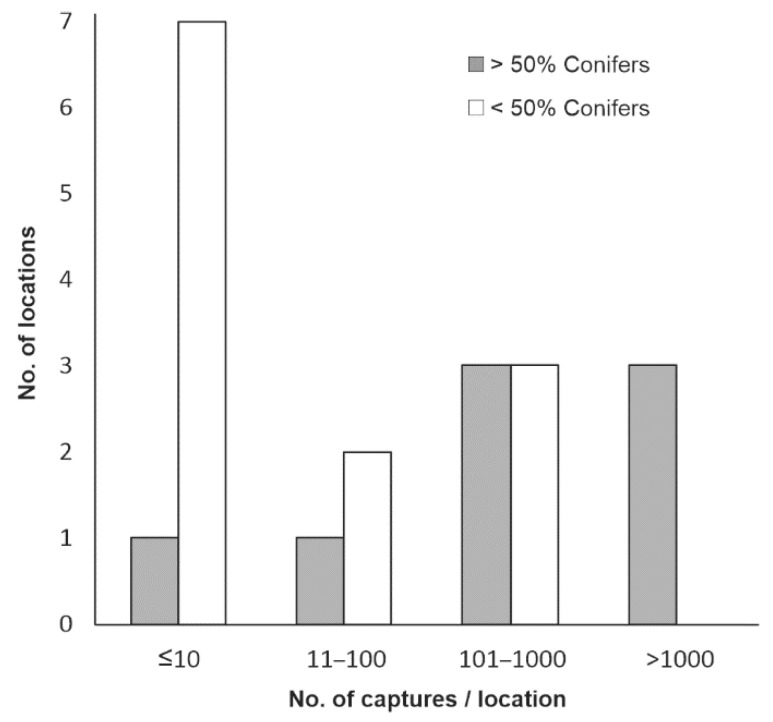
NBB capture variation according to tree stand composition.

**Figure 4 insects-13-00621-f004:**
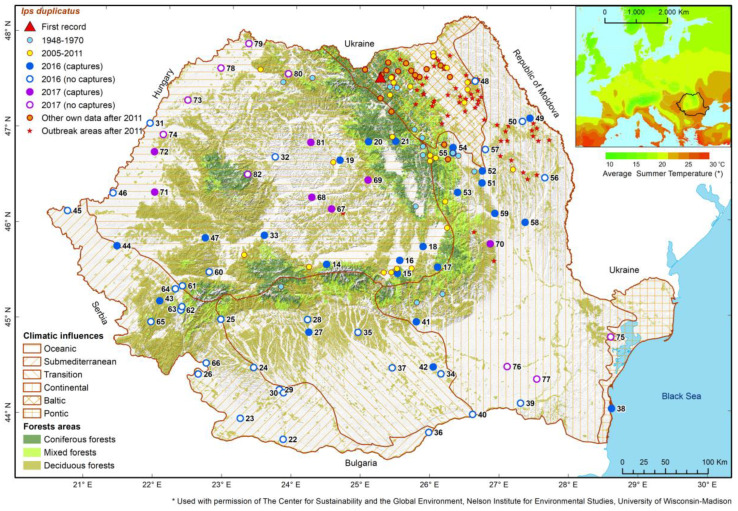
Geographic distribution of places where NBB presence was detected between 1948 and 2017. (The correspondences between point numbers and names of locations are given in Appendix A).

**Figure 5 insects-13-00621-f005:**
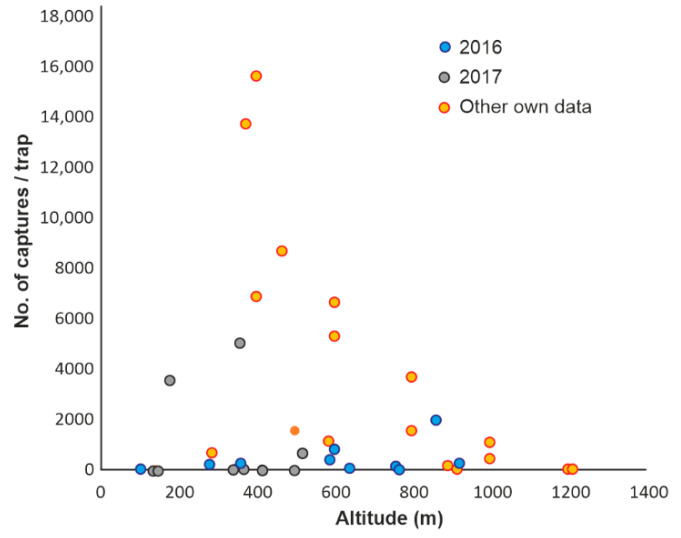
Variation of NBB captures with altitude in different years. Only studies that were carried out throughout the vegetation season and used pheromone lures for NBB were considered.

**Figure 6 insects-13-00621-f006:**
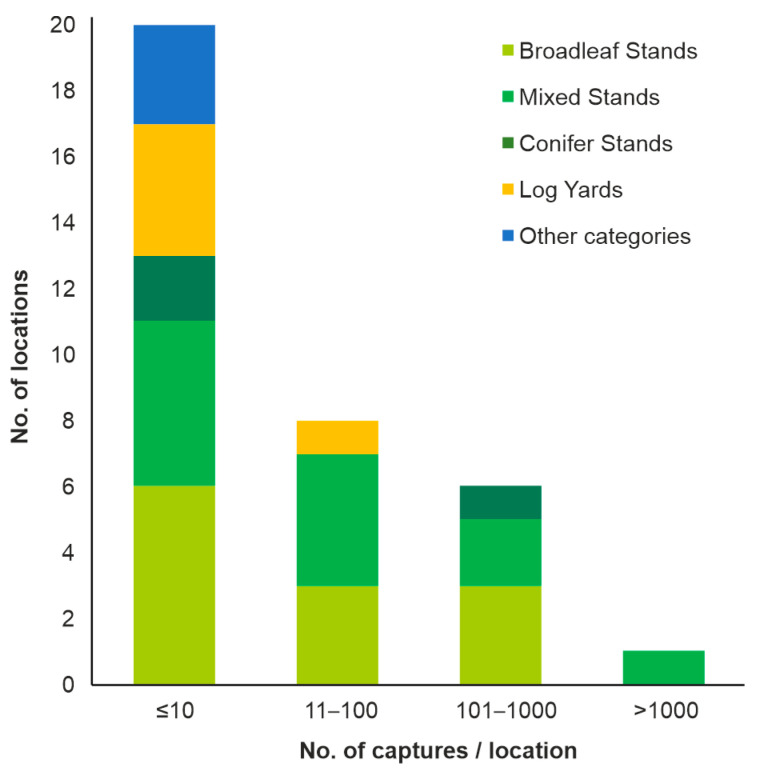
Number of locations and habitats with different BSB capture sizes.

**Figure 7 insects-13-00621-f007:**
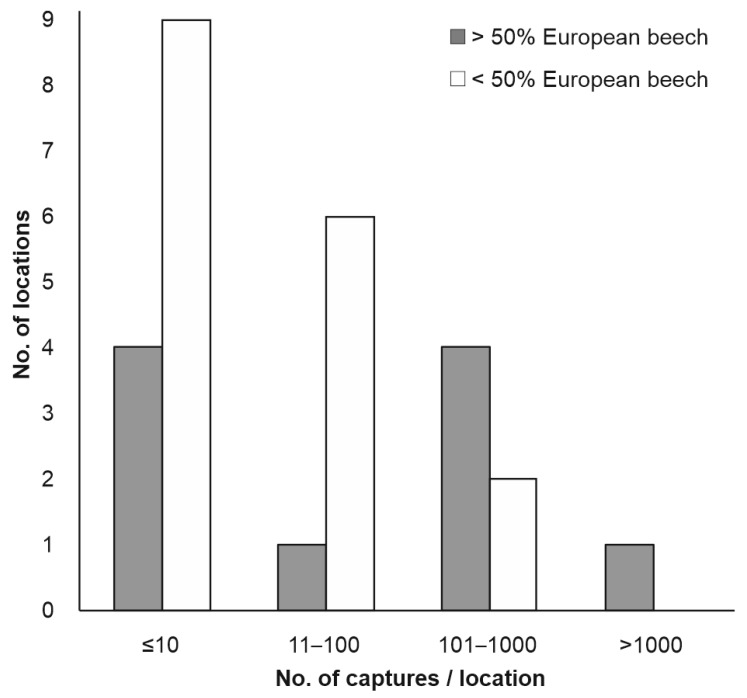
BSB capture variation according to tree stand composition.

**Figure 8 insects-13-00621-f008:**
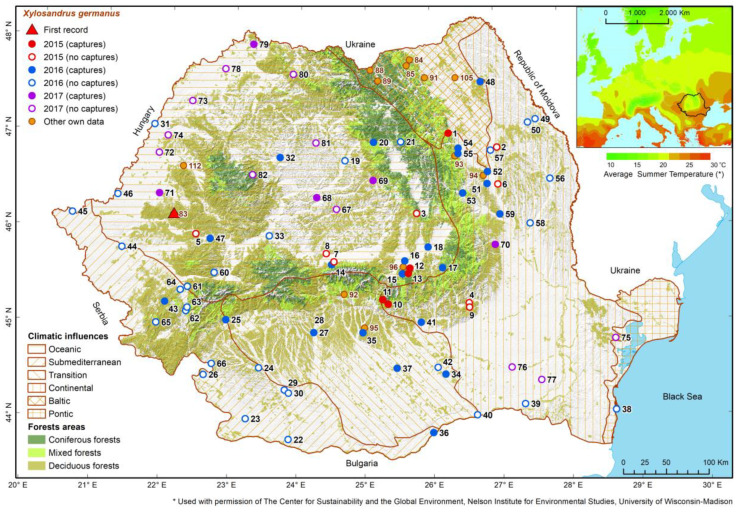
Geographic distribution of places where the BSB presence was detected between 2009 and 2017. (The correspondences between point numbers and names of locations are given in Appendix A).

**Figure 9 insects-13-00621-f009:**
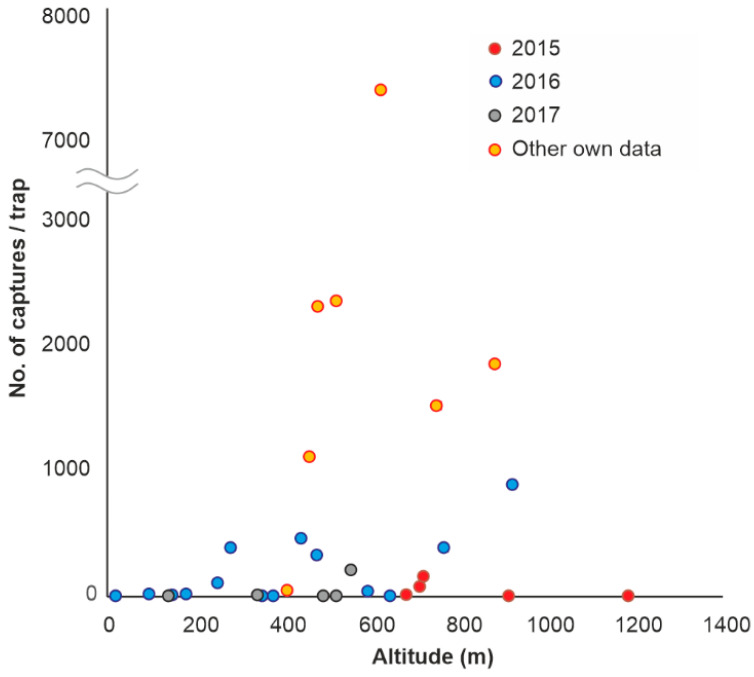
Variation of BSB captures with altitude in different years. Mainly studies carried out throughout the vegetation season were taken into account.

**Figure 10 insects-13-00621-f010:**
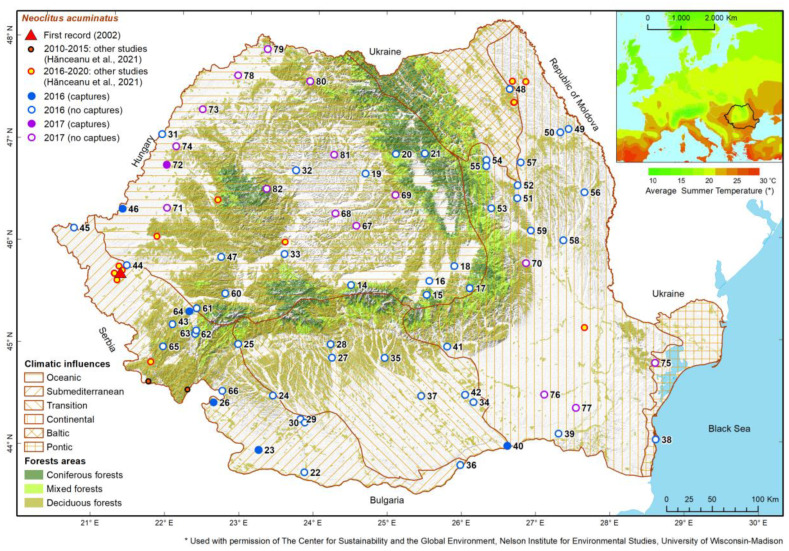
Geographic distribution of places where RHAB presence was detected between 2002 and 2020, based on data from own and other studies [1]. (The correspondences between point numbers in our study and names of locations are given in Appendix A).

**Table 1 insects-13-00621-t001:** The lure types used in the study and their characteristics.

Target Species	Lure Type	Lure Composition	Release Rate (mg d^−1^) at 20 °C	Functioning Period (Weeks)
*I. duplicatus*	AtraDUP^®^ (D)	E-myrcenol, Ipsdienol, 2-methyl-3buten-2-ol	28	5–6
*X. germanus*	Ethanol (E)	Ethyl alcohol 96%	430	6
*N. acuminatus*	Ethanol + alpha-pinene (E + P)	Ethyl alcohol 96% (-)-α-pinene	430 1300	6 1
Other species	AtraTYP^®^ (T)	2-methyl-3-buten-2-ol (-)-*cis*-Verbenol	31	6–8
	AtraLINEA^®^ (L)	2-methyl-3-buten-2-ol Ethyl alcohol, (S)-α-Pinene, Lineatin	30	12

**Table 2 insects-13-00621-t002:** The number of locations with different types of traps and lures.

Year	Number of Locations	Lures ^1^	Number and Type of Traps ^2^ in Each Location
2015	13	E	3 PTd
2016	29	D, E, E + P, L, T	4 PTw + 1 BT
	12	E, E + P, L	2 PTw + 1 BT
	4	E, P, L	2 PTw + 1 BT
	3	D, P, L, T	4 PTw
	2	D, E + P, T	3 PTw
	1	D, E, P, T	3 PTw + 1 BT
	1	D, E, P, L, T	4 PTw + 1 BT
	1	E, E + P	1 PTw + 2 BT
2017	15	D, E, E + P, L, T	5 PTw
	1	D, E, E + P, T	4 PTw

^1^ Lure type: E—Ethanol, D—AtraDUP, E+P—Ethanol + Alpha-Pinene, L—AtraLINEA, T—AtraTYP. ^2^ Trap type: PTd—dry panel trap, PTw—wet panel trap, BT—bottle trap.

**Table 3 insects-13-00621-t003:** Distribution of locations where pheromone traps for NBB were set up and its presence detected.

Captures of *I. duplicatus*	Conifer Stands	Mixed Stands	Broadleaf Stands	Wood Yards	Airports	Other Categories
Yes	7	9	4	6	2	2
No	2	2	9	1	2	6

**Table 4 insects-13-00621-t004:** Distribution of locations where kairomone traps for BSB were set up and the species’ presence detected.

Captures of *X. germanus*	Conifer Stands	Mixed Stands	Broadleaf Stands	Wood Yards	Airports	Other Categories
Yes	3	12	12	5	1	2
No	1	10	20	7	3	3

## Data Availability

On reasonable request, derived data supporting the findings of this study are available from the corresponding author.

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
