# Peer review of "Geographical Distribution of Three Forest Invasive Beetle Species in Romania"

_insects, 2022, doi:10.3390/insects13070621_

Round 1
Reviewer 1 Report
The authors of this study present a very large amount of detailed survey data from baited traps, documenting the geographical distribution (as measured by trap catch) of three species of non-native and potentially invasive bark-, ambrosia-, longhorn beetles within Romania. The data as presented in the main paper and especially in the tables of supplementary data are very extensive and are relevant to forest managers in Romania and neighboring countries. The authors also looked at climatic data and data on wood imports gathered from various sources in an effort to explain the geographical distribution pattern they observed from the trap capture survey data. The English was very good for the most part with only some minor suggestions made to improve flow and clarity.
However, I do not find the manuscript acceptable for publication in its present state and I am not sure if it can be acceptable even after major revision. My main reasons for this rather negative review are as follows: 1) while the survey data on distribution of these exotic beetles is relevant to foresters and forest managers in Romania, none are new country records for Romania and as such, do not make a major contributions to the species worldwide distribution; 2) the authors have speculated a great deal about the possible factors affecting the geographical distribution they observed, e.g., climate change, host tree distribution, and imports of infested wood from other countries, but it is difficult to understand what is speculation and what is backed up by data in this study. The data generated in this study consists mainly of trap catch survey data, and some climate data from published sources. While the discussion of possible factors influencing the distribution of these species in Romania is quite well written and interesting, I think it tends to go beyond what is supported by the data collected at times; 3) the attractants used in the traps for survey of Neoclytus acuminatus were not appropriate; the authors acknowledge this limitation when discussing their results but the distribution of this species would likely be much greater than reported here if traps had been baited if Fluon-treated traps had been baited with the aggregation pheromone syn-2,3-hexanediols, which has been commercially available for at least a decade.
I have also made some minor comments in an annotated pdf of the manuscript. I attempted to attach this file to the review twice but each time was unsuccessful ("unknown error") so iIwill email that pdf directly to the Editor.

Author Response
We would like to thank the Reviewer 1 for the reviewing our manuscript and for the valuable suggestions provided.
The comments and suggestions were relevant and have helped us improving greatly the manuscript. We agreed most changes suggested for the manuscript. We are therefore pleased to send to you the revised manuscript, hopping that we successfully addressed all the comments.
Review form
The authors of this study present a very large amount of detailed survey data from baited traps, documenting the geographical distribution (as measured by trap catch) of three species of non-native and potentially invasive bark-, ambrosia-, longhorn beetles within Romania. The data as presented in the main paper and especially in the tables of supplementary data are very extensive and are relevant to forest managers in Romania and neighboring countries. The authors also looked at climatic data and data on wood imports gathered from various sources in an effort to explain the geographical distribution pattern they observed from the trap capture survey data. The English was very good for the most part with only some minor suggestions made to improve flow and clarity.
However, I do not find the manuscript acceptable for publication in its present state and I am not sure if it can be acceptable even after major revision. My main reasons for this rather negative review are as follows: 1) while the survey data on distribution of these exotic beetles is relevant to foresters and forest managers in Romania, none are new country records for Romania and as such, do not make a major contributions to the species worldwide distribution; 2) the authors have speculated a great deal about the possible factors affecting the geographical distribution they observed, e.g., climate change, host tree distribution, and imports of infested wood from other countries, but it is difficult to understand what is speculation and what is backed up by data in this study. The data generated in this study consists mainly of trap catch survey data, and some climate data from published sources. While the discussion of possible factors influencing the distribution of these species in Romania is quite well written and interesting, I think it tends to go beyond what is supported by the data collected at times; 3) the attractants used in the traps for survey of Neoclytus were not appropriate; the authors acknowledge this limitation when discussing their results but the distribution of this species would likely be much greater than reported here if traps had been baited if Fluon-treated traps had been baited with the aggregation pheromone syn-2,3-hexanediols, which has been commercially available for at least a decade.
Answers on the three main issues reported:
- Indeed, none of the three species are reported for the first time in Romania and - consequently - the paper does not make a major contribution to their global distribution. However, the publication of this data in a prestigious journal aims to ensure an increased visibility of this information, for inclusion in international databases. If we consult, for example, the Invasive Species Compendium (www.cabi.org), to see what is the global distribution of the species Ips duplicatus, which was updated on May 12, 2022, we will see that for Romania it is mentioned only that the species is present, without further details, although preliminary results on this issue have already been published (see Duduman et al., 2011). The situation is similar in the case of the species Xylosandrus germanus, a species for which it is mentioned as a bibliographic reference EPPO (2022), although information about this species was published in a journal ISI, namely Annals of Forest Research (Olenici et al., 2014).
- The distribution of the three species in Romania, as shown on the map, is based on field observations. Regarding the factors that could have affected the distribution of these species, we sought to document with data the changes in the distribution of spruce in Romania and climate change in recent decades, which contributed to the distribution of I. duplicatus outside the natural area of ​​spruce and to the development of the first outbreak in our country. We argued in the Discussions chapter how these factors contributed to the current situation. Concerning the changes regarding the imports of round timber, these were taken into account especially in order to understand the appearance - at an interval of only two years - of quite large populations of germanus in two distinct areas of the country, in the west and the northeast of the country, a fact that suggests multiple introductions in different places. Indeed, this is speculation, but multiple international research has highlighted the role of timber imports in introducing such pests into areas where they were not previously.
- Indeed, the attractants used to detect the presence of the species acuminatus were not the most suitable, but we honestly recognize the limitations of the method used for this species and - for the reader to have a more accurate picture of the real situation - on the distribution map of this species we have completed the information obtained by us with the information obtained by other researchers and which are quoted accordingly. Thanks for the suggestion to use the specific aggregative pheromone and we will take it into account in future research.
Text of the paper
Line 371-372 – Reviewer: this relatively low capture rate was very likely greatly influenced by the poor attractants used to bait the traps, and this should be acknowledged.
Lines 627-631 – Reviewer: yet Miller et al. captured more N. acuminatus in traps placed at 1.5 m than in traps placed in the canopy.
Response: We added on the line 631 "Yet Miller et al. (2020) captured more N. acuminatus in traps placed at 1.5 m than in traps placed in the canopy" and the new reference in the list of references.
Reviewer 2 Report
The article deals with the distribution of three invasive beetle species in Romania. Most of the data obtained are from the authors' research. It is a pity that the authors did not focus more on trapping on the Black Sea coast, where they had only two sites. It is the ports that are the entry point for invasive beetles, as confirmed by data from Italy and France.
I recommend the paper for acceptance after minor revisions.
line 57 - citation [23] inconsistent with Scolytinae, but with Cerambycidae - correct
line 302 - deleted (Bogdan, 1983)
Figure 1, 4 and 8 - replace the figures with better quality ones, the text in the figures is blurry and hard to read
In the introduction explain why Ips duplicatus is an invasive bark beetle in Romania when it was detected in 1948, but not Trypodendron laeve, which was only detected in Romania in 2008, when both species are native to the northern Palaearctic
Author Response
We would like to thank the Reviewer 2 for the reviewing our manuscript and for the valuable suggestions provided.
The comments and suggestions were relevant and have helped us improving the manuscript. We agreed most changes suggested for the manuscript. We are therefore pleased to send to you the revised manuscript, hopping that we successfully addressed all the comments.
Review form
The article deals with the distribution of three invasive beetle species in Romania. Most of the data obtained are from the authors' research. It is a pity that the authors did not focus more on trapping on the Black Sea coast, where they had only two sites. It is the ports that are the entry point for invasive beetles, as confirmed by data from Italy and France.
Response: In the last decades, Romania imported relatively little wood and it came mostly from Europe, the transport being mainly made by rail. However, various pests can enter the country with wooden packaging or pallets. We considered this aspect when we chose the port of Constanta as one of the study sites. Of course, a higher number of sites is always desirable, especially in areas where the entry of new pests is most likely and we will take this suggestion into account in future research.
Text of the paper
Line 57 – Reviewer: citation [23] inconsistent with Scolytinae, but with Cerambycidae – correct
Response: In the list of references, another bibliographical reference was mistakenly included. The correct one is: "Faccoli, M., Campo, G., Perrotta, G., Rassati, D., 2016. Two newly introduced tropical bark and ambrosia beetles (Coleoptera: Curculionidae, Scolytinae) damaging figs (Ficus carica) in southern Italy. Zootaxa 4138 (1): 189–194." and I fixed this mistake.
"Faccoli, M., Campo, G., Perrotta, G., Rassati, D., 2016. Two newly introduced tropical bark and ambrosia beetles (Coleoptera: Curculionidae, Scolytinae) damaging figs (Ficus carica) in southern Italy. Zootaxa 4138 (1): 189–194."
Line 302 – Reviewer: deleted (Bogdan, 1983)
Response: We deleted "(Bogdan, 1983)".
Reviewer: Figure 1, 4 and 8 - replace the figures with better quality ones, the text in the figures is blurry and hard to read.
Response: We improved the quality of the figures, as suggested.
Reviewer: In the introduction explain why Ips duplicatus is an invasive bark beetle in Romania when it was detected in 1948, but not Trypodendron laeve, which was only detected in Romania in 2008, when both species are native to the northern Palaearctic
Response: The data presented in the Introduction clearly show that Ips duplicatus is of boreal origin and that it gradually entered Central Europe during the 20th century. There is evidence that the expansion of the distribution area was done with the help of man, respectively through the transport of infested wood from northern Europe to Central Europe. Towards the end of the 20th century and the beginning of the 21st century, it also developed the first outbreaks in Central Europe and Romania, respectively, thus causing important damages. All these elements indicate that we are dealing with an invasive species. In contrast, Trypodendron laeve seems to have been present in the mountainous area of Central Europe and in Romania for a very long time, but it was overlooked because its taxonomic status was confuse. Now, its taxonomic status is clear, but its origin remains a topic of discussion and perhaps genetic analysis will help clarify this issue.
Reviewer 3 Report
Line 71: That region is regarded as a native area of occurrence for I. duplicatus [48]. Compare this statement with reference [60].
Line 85-89: was the main import pathway to Central and Western Europe (un-debarked timber from Russia and Baltic countries) confirmed also genetically?
Line 104: trees or cubic meters?
Line 118-120: better separate to two sentences
Line 125: specify which conifers mainly
Line 203: Figure 1 The pie charts on the map should have been slightly larger. On the A4 page are poorly visible.
Chapter 2.1 Study Areas does not clearly explain the selection pattern of study localities—regular, random, strata.
Line 242-246: The time of the trap installation is connected to the elevation and swarming period of the species. I'm suggesting adding information about elevations to table 2.
Line 292-296: No statistical tests are visible in the whole results
Line 332: the colour palette in the legend Figure 2 should be larger, in the size A4 is not visible
Line 391: delete dot at the beginning
Line 493: genus name is abbreviated contrary to the other two chapters' title
Line 711: M.J. not M.a.J.
Line 723: two times doi
Line 731-732: non-standard reference
Line 739: behind year ; should be ,
Line 751: unify letter „e“
Line 850: range of pages is missing
Line 904: range of pages is missing
Line 1054: range of pages is missing
Line 1083: number 888. is redundant
Author Response
We would like to thank the Reviewer 3 for the reviewing our manuscript and for the valuable suggestions provided.
The comments and suggestions were relevant and have helped us improving greatly the manuscript. We agreed most changes suggested for the manuscript. We are therefore pleased to send to you the revised manuscript, hopping that we successfully addressed all the comments.
Text of the paper
Line 71 - Reviewer: That region is regarded as a native area of occurrence for I. duplicatus [48]. Compare this statement with reference [60].
Response: To be more clear, we changed the text: "That region is regarded as the southern border of the native area of occurrence for I. duplicatus [48,60]".
Lines 85-89 - Reviewer: was the main import pathway to Central and Western Europe (un-debarked timber from Russia and Baltic countries) confirmed also genetically?
Response: We inserted in the line 88 the sentence: "This introduction pathway was confirmed by genetic analyses (Lakatos et al., 2007)" and the bibliographic reference in the list of references.
Line 104 - Reviewer: trees or cubic meters?
Response: The value represent the number of trees, as reported by the forest administration. However, to be mor clear, we added on the line 104 "… trees (0.4 million cubic meters) …"
Line 118-120 - Reviewer: better separate to two sentences
Response: We separated the phrase, as suggested. The new version of text is: "Consequently, it does not depend on tree species itself, but on temperature and the humidity which ensure fungus growth. It is a very polyphagous species, mainly colonizing broadleaves, but also conifers [79]".
Line 125 - Reviewer: specify which conifers mainly
Response: We replaced the word "conifers" with "Norway spruce and silver fir".
Line 203: Figure 1 The pie charts on the map should have been slightly larger. On the A4 page are poorly visible.
Response: We modified the pie charts on the map, as suggested.
Chapter 2.1 - Reviewer: Study Areas does not clearly explain the selection pattern of study localities—regular, random, strata.
Response: We have completed on line 175 and the new version of the text is: "The study locations were chosen randomly taking into account …".
Line 242-246: The time of the trap installation is connected to the elevation and swarming period of the species. I'm suggesting adding information about elevations to table 2.
Response: Information about elevation is presented for each study site in the Table S1 (Supplementary material).
Line 292-296 - Reviewer: No statistical tests are visible in the whole results
Response: Only climatic data, which are presented in figures S1-S4 (Supplementary material), were subjected to statistical testing. This is the reason why no statistical test is mentioned in the Results chapter.
Line 332 - Reviewer: the colour palette in the legend Figure 2 should be larger, in the size A4 is not visible
Response: We modified the figure, as suggested.
Line 391 Reviewer : delete dot at the beginning
Response: We deleted the dot.
Line 493 - Reviewer: genus name is abbreviated contrary to the other two chapters' title
Response: We have rectified the text. The chapter’s title is "Xylosandrus germanus".
Line 711 - Reviewer: M.J. not M.a.J.
Response: We changed the text, as suggested.
Line 723 - Reviewer: two times doi
Response: We deleted one doi.
Line 731-732: non-standard reference
Response: In the first version of the manuscrips we gave the citation as suggested in that book. However, in the revised form we changed the reference, as follows: Reid, W.V.; Mooney, H.A.; Cropper, A.; Capistrano, D.; Carpenter, S.R.; Chopra, K.; Dasgupta, P.; Dietz, T.; Duraiappah, A.K.; Hassan, R.; Kasperson, R.; Leemans, R.; May, R.M.; McMichael, A.J.; Pingali, P.; Samper, C.; Scholes, R.; Watson, R.T.; Zakri, A.H.; ... Zurek, M.B. Ecosystems and human well-being - Synthesis: A Report of the Millennium Ecosystem Assessment; Island Press: Washington, DC., 2005; pp. 137.
Line 739 - Reviewer: behind year ; should be ,
Response: We replaced ; with ,
Line 751- Reviewer: unify letter „e“
Response: The new version is E2264-E2273.
Line 850 - Reviewer: range of pages is missing
Response: the article is only one page long (the page number 100 in the journal).
Line 904 - Reviewer: range of pages is missing
Response: We completed the reference with the number of pages (100).
Line 1054 - Reviewer: range of pages is missing
Response: We completed the reference with the number of pages (129).
Line 1083 - Reviewer: number 888. is redundant
We deleted the number 888.
Round 2
Reviewer 1 Report
Please make a stronger statement regarding the weakness of the survey data and very likely underestimated distribution of Neoclytus acuminatus in Romania due to the use of ethanol and alpha-pinene as trap lures instead of the species aggregation pheromone. Had you used traps baited with syn-,2,3-hexanediols plus ethanol, the probability of detecting N. acuminatus would have been much, much greater, and consequently, several of your survey locations with negative data may actually have established populations. I strongly suggest you include some of these pheromone + ethanol baited traps in new surveys for this longhorn beetle in Romania to get a more accurate estimate of its distribution. I suggest thefollowing sentence or one similar be inserted on lines 614-617 to replace the one that is currently there: "However, the actual distribution of RHAB in Romania may be much greater than suggested by our trapping survey data because the lures we used (ethanol, alpha-pinene) are much less effective than the species aggregation pheromone, syn-2,3-hexanediols (183, 184). For example, in Georgia USA, funnel traps baited with the combination of syn-2,3-hexanediols plus ethanol had mean captures of about 100 RHAB compared to zero or nearly zero catches in traps baited with ethanol alone (Miller et al. 2015)" https://doi.org/10.1093/jee/tov220
Author Response
Dear Reviewer 1,
We made all the requested changes.
Thanks you very much for the constructive review of this manuscript.